# How Work–Nonwork Conflict Affects Remote Workers’ General Health in China: A Self-Regulation Theory Perspective

**DOI:** 10.3390/ijerph20021337

**Published:** 2023-01-11

**Authors:** Yanwei Shi, Dan Li, Zhiqing E. Zhou, Hui Zhang, Zhuang She, Xi Yuan

**Affiliations:** 1Department of Human Resource Management, Shanghai Normal University, Shanghai 200234, China; 2College Student Mental Health Education and Consultation Center, Hainan Medical University, Haikou 570216, China; 3Department of Psychology, Baruch College and The Graduate Center, City University of New York, New York, NY 10010, USA; 4School of Sociology, Huazhong University of Science and Technology, Wuhan 430074, China; 5Shanghai Key Laboratory of Mental Health and Psychological Crisis Intervention, Affiliated Mental Health Center (ECNU), School of Psychology and Cognitive Science, East China Normal University, Shanghai 200062, China

**Keywords:** work–nonwork conflict, general health, self-control capacity impairment, perceived boundary control, remote worker, self-regulation theory

## Abstract

Difficulty in balancing the demands of work and nonwork has been shown to be associated with lower physical and psychological health. Grounded on the self-regulation theory, we examined the effect of work–nonwork conflict on general health among employees who transitioned to remote work (remote workers), and we tested whether this association was mediated by impaired self-control capacity. The study further examined the perceived boundary control as a moderator of these associations. We collected two waves of questionnaire data with a one-month interval from 461 remote workers, and the results of regression-based analyses revealed that work–nonwork conflict was negatively related to remote workers’ general health through increased self-control capacity impairment. In addition, this indirect effect was weaker for remote workers with higher perceived boundary control than those with lower perceived boundary control. These findings expand our understanding of remote workers’ work–nonwork conflict and have practical implications for promoting the general health of remote workers who are experiencing work–nonwork conflict.

## 1. Introduction

The rapidly increasing availability of technological tools is changing where and how we work [1], and these information and communication technologies have led to an increase in the number of remote workers [2], especially after the COVID-19 pandemic [3]. For example, it was estimated 558 million employees globally worked from home during the second quarter of 2020, accounting for 17.4% of the world’s workforce [4]. Although home-based remote work may decrease commuting time and day-to-day office demands, it can also have negative implications for remote employees [5], such as a higher level of work–nonwork conflict. Indeed, the level of work–nonwork conflict has increased for remote workers after the COVID-19 pandemic [6,7], largely because remote work blurs the boundary between employees’ work and nonwork roles. When these roles are highly integrated, employees may experience uncertainty or difficulty in preventing work from affecting other aspects of their lives [8].

Researchers have also started to examine how work–nonwork conflict might affect remote workers’ work and nonwork outcomes. For example, remote workers’ experience of work–nonwork conflict has been linked with higher job anxiety [9], lower work engagement [10], and lower job performance [11], as well as lower positive affect [10] and more burnout [12]. While the latter two studies suggest that remote workers are likely to experience impaired health due to increased work–family conflict, they only provide a narrow focus on individual indicators of employee health, and the current study aims to extend this line of research by examining the effect of work–nonwork conflict on remote workers’ general health. General health reflects an individual’s perceptions of their physical symptoms, anxiety symptoms, sleep disturbance, social functioning, and depression symptoms [13]. Employees’ general health is important not just for the employees themselves [14] but also for organizations [15]. Given the importance of individual general health [14,15], exploring the effect of work–nonwork conflict on remote workers’ general health, and potential underlying mechanisms of this effect, can facilitate our understanding of how to better promote general health for remote workers.

To our knowledge, only one study, conducted in Germany during the COVID-19 pandemic, has examined the relationship between work–nonwork conflict and remote workers’ general health [9]. The study did not find a significant relationship between work–family conflict (one aspect of work–nonwork conflict) and general health. This result might be due to Lange and Kayser’s use of a single item from the Minimum European Health Module to assess self-perceived general health. Meanwhile, the study focused on work–family conflict but not work–self conflict (the other aspect of work–nonwork conflict). To address these gaps, in the present study, we used a more comprehensive measure (i.e., a well-validated 10-item self-report measure of general health that provides a broader assessment of mental and physical health [13]) to estimate the link between work–nonwork conflict (including work–family conflict and work–self conflict) and general health.

There is also a need to examine the process through which work–nonwork conflict influences remote workers’ general health. Identifying potential mechanisms will help us better understand how to disrupt this association to promote better general health. Self-regulation theory [16] suggests that self-control capacity impairment might be a mediator in the link between work–nonwork conflict and remote workers’ general health. A general assumption of self-regulation theory is that a person’s self-regulation resources, or self-control capacity, are limited. When people have fewer self-regulation resources, or impaired self-control capacity, they are less able to regulate their behavior, attention, and emotions, or to resist temptations [16]. We assume that the demands of work–nonwork conflict reduce these resources because individuals’ self-regulatory resources can be depleted when they try to solve problems [17]. Thus, work–nonwork conflict, as a job-related stressor, may be a demand that consumes remote workers’ self-regulation resources and leads to impaired self-control capacity. Remote workers would then have fewer self-regulation resources (lower self-control capacity) to engage in health-related behaviors such as maintaining an exercise regimen, healthy eating, or healthy sleep patterns. Therefore, grounded on self-regulation theory, we argue that impaired self-control capacity may mediate the relationship between work–nonwork conflict and remote workers’ general health.

According to self-regulation theory, person-based characteristics play an important role in mitigating the process by which self-regulatory resources are lost. People with a larger pool of self-regulation resources are less likely to be affected in this process [16,18]. Thus, the positive relationship between work–nonwork family conflict and self-control capacity impairment may be weaker when remote workers have more self-regulation resources. In the present study, we chose perceived boundary control as an indicator of these resources.

Perceived boundary control is the perception that one “can control the timing, frequency, and direction” of mental and physical transitions between work and family responsibilities [19]. Employees who perceive themselves as having high autonomy in managing work and nonwork roles—that is, those who have high perceived boundary control—also have other psychological resources [19,20,21] (e.g., psychological job control and self-identity) and a lower risk of resource depletion (e.g., emotional exhaustion). According to self-regulation theory [16,18], remote workers’ perception of autonomy in maintaining boundaries could weaken the relationship between work–nonwork conflict and impairment in the capacity for self-control. Thus, we tested perceived boundary control as a moderator that decreases the indirect effect of work–nonwork conflict on remote workers’ general health via impairments in the capacity for self-control (see Figure 1). We used a multi-wave research design at two time points with a one-month interval to collect data from full-time employees in China who worked remotely.

This study makes three contributions to the literature. First, we extend the limited research on the potential effects of work–nonwork conflict on remote workers’ well-being by examining its effects on general health. Given the dramatic increase in the proportion of remote workers in the workforce [3] and their experiences of work–nonwork conflict [7], investigating the relationship between work–nonwork conflict and remote workers’ general health can facilitate our understanding of how to better promote these workers’ general health. Second, this is the first study to test self-control capacity impairment as a potential mediator of the association between work–nonwork conflict and general health. The results will provide a deeper understanding of how work–nonwork conflict affects remote workers’ general health. Finally, by investigating perceived boundary control as a potential moderator, we can better understand how to mitigate the indirect effect of work–nonwork conflict on general health through self-control capacity impairment [19].

### 1.1. Theoretical Framework

The self-regulation theory [16,22] provides the theoretical foundation for the current study’s focus on how work–nonwork conflict may impair self-control capacity, with subsequent damage to remote workers’ general health. One key proposition of the theory is that self-regulatory resources are limited [16]. Self-regulatory resources constitute “the internal resources available to inhibit, override, or alter responses that may arise as a result of physiological processes, habit, learning, or the press of the situation” [23]. When people experience stressful situations, they may exert self-control, use these self-regulatory resources, and experience self-regulatory resource depletion; consequently, they are likely to suffer from self-control capacity impairment [24]. When people experience impaired self-control capacity, they will have fewer self-regulatory resources to engage in other behaviors that also require self-control such as engagement in health-related behaviors.

Applying self-regulation theory to our study, we assert that work–nonwork conflict as a stressful experience impairs remote workers’ self-control capacity and leaves workers with fewer self-regulatory resources; subsequently, they are likely to engage in fewer health-related behaviors and experience poorer general health.

### 1.2. Hypotheses

Work–nonwork conflict occurs when work role demands interfere with nonwork role demands [25], and the two types of demands compete for resources [17,26]. Research in this area has focused on two pivotal aspects of work–nonwork conflict: work–family conflict and work–self conflict [27]. According to self-regulation theory [16], work–nonwork conflict can consume resources for self-regulation [16,17] and thus impair the capacity for self-control. Thus, we propose that remote workers’ self-regulatory resources are taxed when they experience more work–nonwork conflict.

Specifically, first, because managing the relationship between work and nonwork roles is an effortful process that involves self-regulation [28], remote workers who experience higher work–nonwork conflict will exert more self-regulation to manage work and nonwork roles. They may thus lose self-regulatory resources [28], leading to impaired self-control capacity. Second, remote workers who experience more work–nonwork conflict show more negative emotion, which depletes self-regulatory resources [22] and leads to impairments in self-control capacity. According to the self-regulation theory [22], remote workers need to use self-control resources to regulate the negative emotions induced by work–nonwork conflict, and thus experience loss of self-regulatory resources. The reduction in these resources in turn impairs self-control capacity. Third, Beal et al. [22] proposed that distraction or interruption would lead to off-task attention, which in turn depletes individuals’ self-regulatory resources. In a similar vein, work–nonwork conflict, as an interruption of the balance between work and nonwork roles, may reduce employees’ focus on work, increase off-task attentional demands, and in turn impair self-control capacity.

Indirect support for these assumptions comes from research on work–family conflict (one aspect of work–nonwork conflict). For example, Dahm [29] reported that work–family conflict depleted employees’ self-regulation resources. Ohu et al. [17] found that work–family conflict was negatively related to employees’ self-regulatory resources. Whereas these studies focused only on work–family conflict, the present study examines the effect of work–nonwork conflict including both work–self conflict and work–family conflict on self-regulation resources. It is critical to include work–self conflict because with the increasing diversity of individuals’ expressed needs, individuals place more value than before on “self” life and reducing work–self conflict [30]. Taken together, the self-regulation theory and the empirical evidence lead us to propose the following:

**Hypothesis** **1 (H1).***Work–nonwork conflict will be positively related to remote workers’ self-control capacity impairment*.

Self-regulation theory also proposes that when self-control capacity is impaired, people will be less likely to function effectively because they have fewer self-regulatory resources to engage in regulated behaviors [18,31]. In the current study, we focused on general health as a nonwork domain outcome and argue that remote workers with impaired self-control capacity will be at risk of poorer general health for two reasons. First, remote workers who experience self-control capacity impairment will have fewer self-regulatory resources to regulate their impulses, and thus will be more likely to engage in impulsive behaviors such as substance abuse and unhealthy eating, resulting in worse general health [32,33].

Second, remote workers with self-control capacity impairment will be less able to maintain health-promoting behaviors such as exercise and sleep hygiene that are directly related to their general health, because these health-promoting behaviors are achieved through an expenditure of self-regulation resources [34,35]. Supporting this possibility, previous research has suggested that self-regulation resources are associated with adherence to health-related behavior [32,35,36]. However, remote workers with impaired self-control capacity would not possess enough self-regulatory resources to face the demands of executing health behaviors that positively affect their general health.

Taken together, the evidence leads us to assert that employees with impaired self-control capacity will experience lower general health due to more unhealthy behaviors and fewer health-promoting behaviors. The results of previous studies provide indirect evidence of this notion. For example, it was found that individuals low in self-control exhibited more unhealthy behaviors such as binge eating and alcohol abuse [37,38], and engage in fewer health-promoting behaviors such as exercise [39]. Further, Hagger [36] found that individuals with low self-control had lower scores on a self-report measure of health. Therefore, we hypothesize the following:

**Hypothesis** **2 (H2).***Self-control capacity impairment will be negatively related to remote workers’ general health*.

Based on self-regulation theory and the aforementioned evidence of the associations among work–nonwork conflict, self-control capacity impairment, and general health, it is possible that self-control capacity impairment mediates the relationship between work–nonwork conflict and general health. Specifically, remote workers with more work–nonwork conflict may consume self-control resources to manage the work and nonwork roles and to regulate negative emotion induced by work–nonwork conflict. They are thus more likely to experience impaired self-control capacity, and they will have fewer self-regulatory resources to control impulses to engage in unhealthy behaviors and to maintain health-promoting behavior, which in turn will decrease general health.

**Hypothesis** **3 (H3).***Self-control capacity impairment will mediate the relationship between work–nonwork conflict and remote workers’ general health*.

Self-regulation theory also suggests that person-based characteristics play an important role in mitigating the loss of self-regulatory resources because some people have a larger pool of self-regulation than others [16,18]. In the current study, we propose that the effect of work–nonwork conflict on self-control capacity impairment may be weaker when remote workers have a higher level of perceived boundary control.

Specifically, higher perceived boundary control can increase self-regulatory resources and reduce the depletion of self-regulatory resources. Remote workers with higher boundary control have more autonomy to determine when and how to adjust the boundaries between work and nonwork [19]. As such, perceived boundary control is considered an important psychological resource [19,20,40]. In addition, perceived boundary control may expand individuals’ pool of self-regulation resources. For example, a previous study found that people with high boundary control perceived themselves as having more psychological job control and a more positive self-identity than those with lower perceived boundary control [19]. Therefore, remote workers who perceive high boundary control are likely to have more self-regulatory resources, and these resources will be sufficient to manage work–nonwork conflict. Hence, compared to remote workers with low perceived boundary control, those with high perceived boundary control will have less impairment of self-control capacity.

In addition, boundary control has the potential to mitigate the effect of work–nonwork conflict on self-control capacity impairment because remote workers with higher boundary control could better cope with work–family conflict. Indirect evidence for this possibility comes from research showing that boundary control at work provides non-telecommuting employees with more effective recovery after work. Boundary control facilitates psychological detachment from work [41] and reduces stress and emotional exhaustion [20]. Thus, even though remote workers experience high work–nonwork conflict, they might cope better with work–nonwork conflict and in turn experience weaker self-regulatory capacity impairment.

Supporting the above argument, previous research proposed that perceived boundary control can enable employees to better cope with job-related stressors and weaken the stressor–strain relationship [42]. Specifically, they found that perceived boundary control moderates the negative relationships between weekly information communication technology demands and negative work rumination. Accordingly, we argue that the relationship between work–nonwork conflict and self-control capacity impairment will be weaker for those with higher perceived boundary control than for those with lower perceived boundary control.

**Hypothesis** **4 (H4).***Perceived boundary control will moderate the first link in the mediated pathway, namely the relationship between work–nonwork conflict and self-control capacity impairment, with the relationship being weaker when perceived boundary control is high*.

According to the self-regulation theory [16], remote workers who are confronted with high work–nonwork conflict are likely to have fewer self-regulation resources, leading to a lower capacity for self-control. This impaired capacity for self-control makes it difficult for remote workers to maintain general health. However, remote workers with high perceived boundary control might show less impairment in self-control capacity, resulting in better general health. Thus, building on Hypotheses 1–4, we propose a moderated mediation model, as follows:

**Hypothesis** **5 (H5).***Perceived boundary control will moderate the indirect effect of work–nonwork conflict on general health through self-control capacity impairment, with the indirect effect being weaker when perceived boundary control is high*.

## 2. Materials and Methods

### 2.1. Participants and Procedure

Our sample constituted 461 full-time employees from different companies in Shanghai, China, who were working remotely due to the outbreak of COVID-19 in the spring of 2022. We used snowball sampling to recruit participants, and data were collected from 1 April to 30 April 2022. To mitigate common method variance, we conducted a two-wave survey with one month between waves [43]. In the first survey (Time 1), we measured the predictor (work–nonwork conflict), the moderator (perceived boundary control), and the control variables. In the second survey (Time 2), we assessed the mediator (self-control capacity impairment) and the outcome (general health).

The research was approved by the Research Ethics Committee of the first author’s institution. Participants entered a numerical code (the last six digits of their 11-digit phone number) on the survey in order to keep the data anonymous and to match the Time 1 and Time 2 data. Each participant received a small monetary reward (CNY 40 = about USD 7). Participants were first informed about the research aim and completion time. They were also informed that the questionnaires were anonymous, and research data would only be used for research purposes. If participants consented to participate, they were then directed to the online data collection interface.

At Time 1, 580 employees participated in the survey. Fifty-nine participants were removed after data collection because they were ineligible for the study (e.g., not working from home due to the lockdown, or frequently working from home prior to the lockdown). Thus, at Time 1, there were 521 usable questionnaires, resulting in a response rate of 89.82%. At Time 2, we only asked the 521 participants who completed usable questionnaires at Time 1 to complete the questionnaires. There were 461 questionnaires submitted (88.48% response rate). Of the final 461 employees who had matched data between Time 1 and Time 2, 280 were female and 181 were male. Their average age was 30.08 years old (SD = 6.06) and the average job tenure was 6.53 years (SD = 6.36). In addition, 169 (36.65%) had at least one child under the age of 18, and 183 (39.69%) had at least one or more elderly persons to care for.

### 2.2. Measures

We used the translation and back-translation method to translate the scales from English into Chinese. English-to-Chinese translation was conducted by a doctoral student who was fluent in both languages. Chinese-to-English back-translation was then conducted by another doctoral student who was fluent in both languages. Discrepancies were resolved through discussion. Unless otherwise noted, the items were rated on a 5-point Likert scale ranging from 1 (strongly disagree) to 5 (strongly agree), with higher scores indicating higher values for the intended variable.

Work–nonwork conflict (T1): Work–nonwork conflict was measured by nine items constituting Netemeyer and Boles’s [44] five-item Work–Family Conflict Scale and Demerouti’s [45] four-item Work–Self Conflict Scale. An example of a work–family conflict item is “Work takes up a lot of my time, which makes it difficult for me to fulfill my family responsibilities”. An example of a work–self conflict item is “I do not fully enjoy my personal interests because I worry about my work”. In our study, the Cronbach’s alpha of the work–nonwork conflict scale was 0.86, and CFA results showed good constructive validity (RMSEA = 0.062, CFI = 0.97, TLI = 0.96).

Perceived boundary control (T1): We used the 4-item scale developed by Kossek and Lautsch [19] to measure perceived boundary control. An example item is “I control whether I am able to keep my work and personal life separate”. In our study, the Cronbach’s alpha of the scale was 0.88, and CFA results showed good constructive validity (RMSEA = 0.010, CFI = 0.99, TLI = 0.99).

Self-control capacity impairment (T2): We used the five-item Self-Control Capacity Scale developed by Twenge et al. [46] to measure employees’ self-control capacity impairment. All items were rated on a 5-point Likert scale ranging from 1 (does not apply at all) to 5 (fully applies), with higher scores indicating a higher level of self-control capacity impairment. An example item is “I feel increasingly less able to focus on anything”. In our study, the Cronbach’s alpha of the work–nonwork conflict scale was 0.86, and the results of CFA showed good constructive validity (RMSEA = 0.086, CFI = 0.98, TLI = 0.97).

General health (T2): Employees’ general health was measured by the 12-item General Health Questionnaire [47]. Half of the 12 items are reverse scored so that a higher score reflects higher general health. Participants rated each item on a 4-point Likert scale ranging from 1 (never) to 4 (always). Example items are “I can enjoy daily life” and “I lose sleep because of worry (reverse scored)”. In our study, the Cronbach’s alpha of the scale was 0.87, and CFA showed relatively good constructive validity (RMSEA = 0.110, CFI = 0.89, TLI = 0.82).

Control variables (T1): Previous studies found that women with jobs who have a child or elder to take care of experience more work–nonwork conflict [48,49]. Thus, we chose gender (male = 1; female = 0), job tenure (1 = less than one year; 2 = one to three years; 3 = more than three years), having at least one child under the age of 18 (yes = 1; no = 0), and whether there was a need for elder care (yes = 1; no = 0) as control variables.

### 2.3. Statistical Analysis

In our study, first, we used SPSS 24 (IBM SPSS Inc., Chicago, IL, USA) to analyze the descriptive statistics, alpha reliabilities for each scale, and correlations between variables. Second, we used AMOS 24 (IBM SPSS Inc., New York, NY, USA to estimate the constructive validity of scales. Finally, we used the SPSS PROCESS macro (IBM SPSS Inc., Chicago, IL, USA) [50] to test our hypotheses.

Specifically, we used PROCESS Model 4 to test Hypotheses 1–3, with self-control capacity impairment as the mediator in the relationship between work–nonwork conflict and general health. Additionally, we used PROCESS Model 7 to test the moderated mediation effect (Hypotheses 4 and 5) with perceived boundary control added as the moderator of the mediation process. Finally, we used PROCESS Model 1 to interpret the interaction between work–nonwork conflict and perceived boundary control in predicting self-control capacity impairment. Bootstrapped bias-corrected confidence intervals (95%) for the indirect effects were generated using 5000 iterations of bootstrapping. An indirect effect is considered significant if zero is not included in its 95% confidence interval (CI).

## 3. Results

### 3.1. Preliminary Analyses

We calculated the means, standard deviations, and correlations among all study variables. Table 1 shows that work–nonwork conflict was positively related to self-control capacity impairment (*r* = 0.42, *p* < 0.001) and negatively related to remote workers’ general health (*r* = −0.44, *p* < 0.001). Self-control capacity impairment was negatively related to general health (*r* = −0.55, *p* < 0.001). These results preliminarily supported our hypotheses.

Confirmatory factor analysis (CFA) showed that a five-factor model that included work–nonwork conflict (work–family conflict and work–self conflict), self-control capacity impairment, perceived boundary control, and general health had acceptable goodness of fit (χ^2^/df = 3.15, CFI = 0.89, IFI = 0.89, RMSEA = 0.07). The one-factor model in which all items of all variables were loaded on a common factor had poor goodness of fit (χ^2^/df = 10.63, CFI = 0.48, IFI = 0.48, RMSEA = 0.15). Together, these results showed that common method variance did not substantially bias the results in this study.

### 3.2. Hypothesis Testing

Hypotheses 1–3 predicted that work–nonwork conflict has effects on remote workers’ general health via self-control capacity impairment. Table 2 shows the results of tests of these hypotheses. Equations (1) and (2) show that after the control variables were entered, work–nonwork conflict predicted self-control capacity impairment (*B* = 0.41, *SE* = 0.04, *p* < 0.001) and negatively predicted general health (*B* = −0.14, *SE* = 0.02, *p* < 0.001). Equation (2) shows that self-control capacity impairment negatively predicted general health (*B* = −0.23, *SE* = 0.02, *p* < 0.001). The indirect effect of work–nonwork conflict on general health via self-control capacity impairment was significant (indirect effect = −0.10, *SE* = 0.02, 95% CI [−0.13, −0.07]). Together, these results supported Hypotheses 1–3.

Hypothesis 4 predicted that perceived boundary control would moderate the relationship between work–nonwork conflict and self-control capacity impairment. Table 3 shows that the interaction between work–nonwork conflict and perceived boundary control significantly predicted self-control capacity impairment (*B* = −0.15, *SE* = 0.06, *p* < 0.01). Further, simple slope analysis showed that the relationship between work–nonwork conflict and self-control capacity impairment was stronger for employees with lower perceived boundary control (1 *SD* below the mean, *B*_simple_ = 0.46, *SE* = 0.07, *p* < 0.001) than for those with higher perceived boundary control (1 *SD* above the mean, *B_simple_* = 0.22, *SE* = 0.07, *p* < 0.01). Figure 2 shows the interaction plot. The results supported Hypothesis 4.

Hypothesis 5 predicted that employee perceived boundary control would moderate the entire mediation process, namely the indirect effect of work–nonwork conflict on general health via self-control capacity impairment. Table 4 shows that the mediation effect was stronger for remote workers with lower perceived boundary control (1 *SD* below the mean, indirect effect = −0.11, *SE* = 0.02, 95% CI [−0.15, −0.07]) than for those with high perceived boundary control (1 SD above the mean, indirect effect = −0.05, *SE* = 0.02, 95% CI [−0.09, −0.02]), and the index of moderated mediation was significant (index = 0.03, *SE* = 0.01, 95% CI [0.01, 0.06]). Thus, Hypothesis 5 was supported.

## 4. Discussion

This study examined the relationship between work–nonwork conflict and the general health of workers who were working remotely due to the COVID-19 pandemic. Based on the self-regulation theory, our study examined the indirect effect of work–nonwork conflict on remote workers’ general health through self-regulation capacity impairment, and the moderating effect of perceived boundary control on this indirect effect. Consistent with the self-regulation theory, we found that work–nonwork conflict impaired self-regulation capacity and in turn predicted poorer general health. We also found that this process was weaker for remote workers who felt that they could determine when, where, and how to navigate conflict between their work and nonwork roles. Among previous studies, only one other study on the relationship between work–nonwork conflict and remote workers’ general health, and our study is based on self-regulation theory, more extensive measures of work–nonwork conflict and general health. In addition, this study contributes to understanding how work–nonwork conflict affects remote workers’ general health and how this link depends on remote workers’ perceived boundary control.

### 4.1. Research Implications

The current study contributes to the literature in several ways. First, the finding that work–nonwork conflict has a negative effect on remote workers’ general health extends the limited research on the relationship between work–nonwork conflict and remote workers’ general health. Specifically, our study enriches the literature by studying both components of the work–nonwork interface, namely work–family conflict and work–self conflict, in relation to remote workers’ general health. Although work–nonwork conflict is typically studied in terms of work–family conflict, work–self conflict is also important given the increasing diversity of individuals’ expressed needs and the increasing value workers place on “self” life and work–self conflict [30,51]. The results also challenge Lange and Kayser’s [9] conclusion that remote workers’ work–family conflict is unrelated to their health.

Second, we found that self-regulation capacity impairment mediated the relationship between work–nonwork conflict and remote workers’ general health. This finding suggests that self-regulation resources play an important role in the process through which work–nonwork conflict influences remote workers’ general health. It also contributes to our understanding of the processes by which remote workers’ work–nonwork conflict might damage their general health. In line with self-regulation theory [18], our findings suggest that when remote workers experience high work–nonwork conflict, they tend to exert more self-control and consume self-regulatory resources, leading to impairment in self-control capacity and consequent damage to general health. Our results are consistent with previous findings in samples of non-remote workers [17,29]. For example, Ohu et al.’s [17] results suggested that employees’ work–family conflict may deplete their self-regulatory resources. Our finding extends previous studies by examining both work–family conflict and work–self conflict in relation to self-regulatory resources. Further, the finding that self-control capacity impairment negatively predicts remote workers’ general health is consistent with self-regulation research in health psychology [35,36].

Finally, we found that high perceived boundary control mitigated the negative effect of work–nonwork conflict on remote workers’ general health through self-regulation capacity impairment. That is, remote workers with higher perceived boundary control were less likely to be influenced by work–nonwork conflict than those with lower perceived boundary control. Our results provide evidence of the moderating role of perceived boundary control in the self-regulation process: remote workers’ perceived boundary control may enrich their pool of self-regulatory resources and help them better deal with work–nonwork conflict. Remote workers with higher perceived boundary control will experience less self-control capacity impairment, and thus will show less damage to their general health. Further, while perceived boundary control may be a key factor in weakening stressors’ impact, it has received little attention in research on work–nonwork boundary management [19]. Our finding adds to the limited literature by showing that perceived boundary control can indeed help employees better navigate between work and nonwork demands and be less affected by the conflict between them.

### 4.2. Practical Implications

The findings have several practical implications for managers and remote workers. First, work–nonwork conflict was negatively related to remote workers’ general health. This finding suggests that remote workers and their managers should take actions to reduce work–nonwork conflict to protect their general health. For example, managers could establish a work–nonwork supportive organizational culture and implement family-friendly policies to create a healthy work environment, in turn decreasing employees’ work–nonwork conflict and enhancing general health [52]. In addition, employees could consider job redesign [53] (e.g., job crafting) or seek social support [54] (e.g., supervisor and spouse support) to help them balance the relationship between work and nonwork.

Second, our results showed that self-control capacity impairment mediates the relationship between work–nonwork conflict and general health. Therefore, managers can explore options to help remote workers increase their capacity for self-control. Researchers have proposed that repeated practice will strengthen self-control [36], and self-control training can improve self-control across domains [55]. Thus, managers can consider offering targeted interventions to reduce remote workers’ self-control impairment.

Finally, our results showed that perceived boundary control buffered the indirect effect of work–nonwork conflict on remote workers’ general health via self-control capacity impairment. This finding suggests that in a working life characterized by blurred boundaries, individuals’ ability to have boundary control can be crucial. Therefore, remote workers are encouraged to find ways to increase their perception of boundary control so that they can have self-regulatory resources (high self-control capacity) to manage work–nonwork conflict. For example, in order to increase work–nonwork boundary control, remote workers could instigate personal boundary management routines such as using the same rituals that are effective in the office [56] (e.g., writing a to-do list for the next day, washing one’s coffee cup), or they can create nano-boundaries to regulate nonwork areas of time and space [57] (e.g., dealing with job-related issues only when children are not present, or only in the home office).

### 4.3. Limitations and Directions for Future Research

There are several limitations of the present study that can be addressed in future research. First, measuring all variables by self-report raises the risk of artificially inflated correlations because of shared method variance. However, our study was designed to mitigate this possibility [58] by using a two-wave design in which different variables were measured at different time points, and empirical tests showed that common method variance did not have a substantial impact on the results. In addition, the self-report approach is appropriate because of the study’s focus on remote workers’ experience of work–nonwork conflict, self-control capacity impairment, and general health. Other people may not be accurate in rating these experiences. Second, our sample only included remote workers in China, and it is unclear whether the findings would generalize to other cultural contexts. Thus, researchers should be cautious in interpreting the results.

Thirdly, while we found evidence to support the mediation pathway, we could not establish temporal precedence or rule out alternative causal pathways. Thus, while our two-wave design is common in work–nonwork conflict research [59,60], it does not allow causal inferences. We encourage future researchers to use a three-wave complete panel design to identify potential causal relationships between work–nonwork conflict and remote workers’ general health.

Finally, our general assumption is that self-control capacity impairment negatively influences health-related behaviors, with subsequent negative effects on general health. However, by measuring general health, not health behaviors, we only tested part of this general assumption. Future research can expand our proposed model by testing health-related behaviors, such as substance abuse and healthy eating, as proximal effects of work–nonwork conflict and as consequent influences on general health.

## 5. Conclusions

Grounded on the self-regulation theory, the current study examined the effect of work–nonwork conflict on general health among remote workers in China and tested whether this association was mediated by impaired self-control capacity. Our study further examined the moderating effect of perceived boundary control on this indirect effect. We found that experiencing higher work–nonwork conflict may have a negative effect on remote workers’ general health by impairing the capacity for self-control. However, this negative effect appears to be buffered by higher perceived boundary control. Together, these findings provide support for our hypothesized moderated mediation model, contribute to our understanding of how work–nonwork conflict affects remote workers’ general health, and suggest a beneficial role of perceived boundary control in managing work–nonwork conflict. Further, given the present empirical evidence, organizations could establish a work–nonwork supportive organizational culture as a way to decrease employees’ work–nonwork conflict and enhance general health. Additionally, employees are encouraged to seek personal social support (e.g., supervisor and spouse support) to help them to balance work and nonwork demands.

## Figures and Tables

**Figure 1 ijerph-20-01337-f001:**
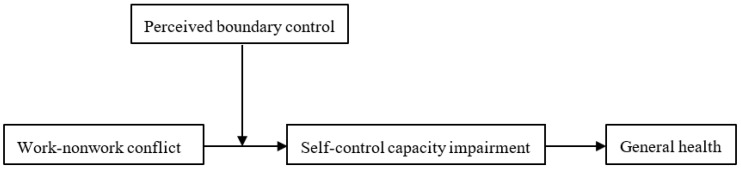
Hypothesized moderated mediation model. The link between work–nonwork conflict and employee general health is mediated by self-control capacity impairment, and the mediation process is weakened by high perceived boundary control.

**Figure 2 ijerph-20-01337-f002:**
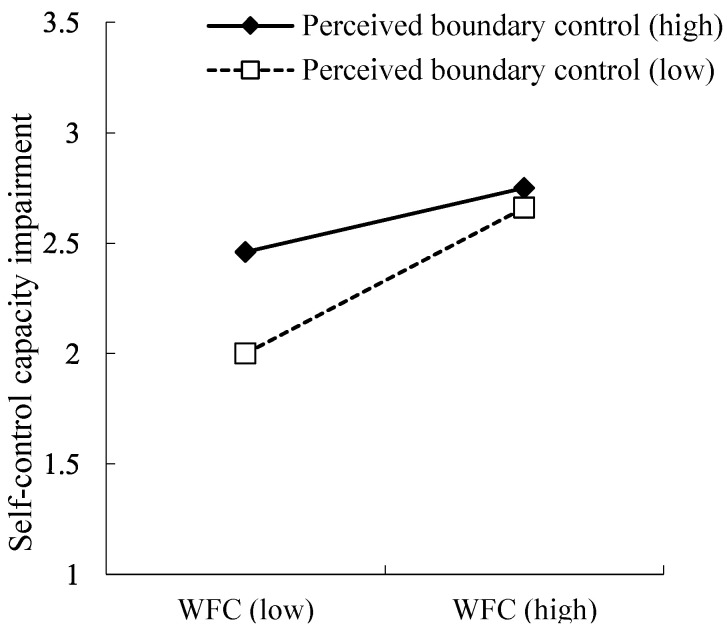
Perceived boundary control moderates the relation between work–family conflict (WFC) and self-control capacity impairment.

**Table 1 ijerph-20-01337-t001:** Means, standard deviations, and correlations among study variables.

Variables	1	2	3	4	5	6	7	8
1. Gender	—							
2. Job tenure	–0.06	—						
3. Child under the age of 18	–0.04	–0.36 ***	—					
4. Elder care	0.07	–0.34 ***	0.43 **	—				
5. Work–nonwork conflict (T_1_)	–0.05	–0.10 *	0.06	–0.09	—			
6. Perceived boundary control (T_1_)	0.03	–0.06	0.04	–0.05	0.41 ***	—		
7. Self-control capacity impairment (T_2_)	–0.05	–0.07	0.01	–0.09 *	0.42 ***	–0.29 ***	—	
8. General health (T_2_)	0.09	0.10 *	–0.06	0.09 *	–0.44 ***	0.38 ***	–0.55 **	—
*M*	—	6.54	—	—	2.61	2.53	2.42	3.02
*SD*	—	6.35	—	—	0.84	0.78	0.86	0.47

*Note*: *N* = 461; T_1_ is the first measurement, and T_2_ is the second measurement; * *p* < 0.05, ** *p* < 0.01, *** *p* < 0.001.

**Table 2 ijerph-20-01337-t002:** Regression results for mediation effect of self-control capacity impairment in the association between work–nonwork conflict and general health.

Predictor	Self-Control Capacity Impairment (T_2_)		General Health (T_2_)	
	Equation (1)		Equation (2)	
	*B*	*SE*	*B*	*SE*
Gender	–0.06	0.08	0.05	0.04
Job tenure	–0.01	0.01	0.01	0.01
Children under the age of 18	0.02	0.09	–0.05	0.04
Elder care	–0.13	0.09	0.06	0.04
Work–nonwork conflict (T_1_)	0.41 ***	0.04	–0.14 ***	0.02
Self-control capacity impairment (T_2_)			–0.23 ***	0.02
*R* ^2^	0.18		0.36	
*F*	19.80 ***		42.61 ***	

*Note*: *N* = 461; T_1_ is the first measurement, and T_2_ is the second measurement; *** *p* < 0.001.

**Table 3 ijerph-20-01337-t003:** Regression results for the moderated meditation effect.

Predictors	Self-Control Capacity Impairment (T_2_)		General Health (T_2_)	
	Equation (1)		Equation (2)	
	*B*	*SE*	*B*	*SE*
Gender	–0.07	0.07	0.05	0.04
Job tenure	–0.01	0.01	0.01	0.01
Children under the age of 18	0.01	0.08	–0.05	0.04
Elder care	–0.12	0.08	0.06	0.04
Work–nonwork conflict (T_1_)	0.34 ***	0.06	–0.14 ***	0.04
Self-control capacity impairment (T_2_)			–0.23	0.03
Perceived boundary control (T_1_)	0.17 **	0.06		
Work–nonwork conflict × perceived boundary control	–0.15 **	0.06		
*R* ^2^	0.21		0.36	
*F*	17.39 ***		34.34 ***	

*Note*: *N* = 461; T_1_ is the first measurement, and T_2_ is the second measurement; ** *p* < 0.01, *** *p* < 0.001.

**Table 4 ijerph-20-01337-t004:** Conditional indirect effects of work–nonwork conflict on general health at different values of perceived boundary control.

	Perceived Boundary Control	Effect	SE_(boot)_	95% CI
	−0.78 (M-SD)	−0.11	0.02	[−0.15, −0.07]
Self-control capacity impairment	0 (M)	−0.08	0.02	[−0.11, −0.05]
	0.78 (M+SD)	−0.05	0.02	[−0.09, −0.02]

## Data Availability

The datasets generated during and/or analyzed during the current study are available from the corresponding author on reasonable request.

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
