# Peer review of "How Work–Nonwork Conflict Affects Remote Workers’ General Health in China: A Self-Regulation Theory Perspective"

_ijerph, 2023, doi:10.3390/ijerph20021337_

Round 1
Reviewer 1 Report
Author should be added the area with location of your study in topic.
Author should be added the hardground of your study setting in introduction part
Author should be added the sampling frame in method part.
Author should be added the measurement validity in method part.
Author should be highlighting the key finding of your work in discussion part.
Author should be added the recommendation from your finding in conclusion.
Reviewer 2 Report
How do the presented theories reflect/related sociodemographis characteristics which are presented in empirical research?
The suggestion is to give some empirical research scheme, whether to show steps of empirical research and hypothesis testing
Can this research be applied to another countries? Are the results important just to China, or other countries can try to lear something?
Giving conclusions – they could reflect hypothesis
